

# A High-resolution Biogeochemical Model (ROMS 3.4 + bio_Fennel) of the East Australian Current System

Carlos Rocha[1], Christopher A Edwards[2], Moninya Roughan[1,3], Paulina Cetina-Heredia[1], Colette Kerry[1]

[1]Coastal and Regional Oceanography Lab, School of Mathematics and Statistics, UNSW Sydney, NSW, 2052, Australia
[2]Department of Ocean Sciences, University of California, 1156 High Street, Santa Cruz, CA 95062, United States
[3]School of Biological, Earth and Environmental Sciences, UNSW Sydney, NSW, 2052, Australia

*Correspondence to*: Carlos Rocha (c.vieirarocha@student.unsw.edu.au)

**Abstract.** Understanding phytoplankton dynamics is critical across a range of topics, spanning from fisheries management to
climate change mitigation. It is particularly interesting in the East Australian Current (EAC) System, as the region's eddy
field strongly conditions nutrient availability and, therefore, phytoplankton growth. Numerical models provide unparalleled
insight into these biogeochemical dynamics. Yet, to date, modelling efforts off southeastern Australia have either targeted
case studies (small spatial and temporal scales) or encompassed the whole EAC System but focused on climate change
effects at the mesoscale (with a spatial resolution of 1/10º). Here we couple a model of the pelagic nitrogen cycle
(bio_Fennel) to a 10-year high-resolution (2.5 - 5 km horizontal) three-dimensional ocean model (ROMS) to resolve both
regional and finer scale biogeochemical processes occurring in the EAC System. We use several statistical metrics to
compare the simulated surface chlorophyll to an ocean colour dataset (Copernicus-GlobColour) for the 2003-2011 period
and show that the model can reproduce the observed phytoplankton surface patterns with a domain-wide *rmse* of
approximately 0.2 mg chla m$^{-3}$ and a correlation coefficient of 0.76. This coupled configuration will provide a much-needed
framework to examine phytoplankton variability in the EAC System providing insight into important ecosystem dynamics
such as regional nutrient supply mechanisms and biogeochemical cycling occurring in EAC eddies.

## 1 Introduction

The basic framework for most marine biogeochemical (BGC) models has been in use for the last few decades (e.g.,
Fasham *et al*., 1990). These models are highly empirical by nature and attempt to describe non-linear processes such as
photosynthesis by phytoplankton, zooplankton grazing or detrital remineralization through idealised formulations that
depend critically on poorly constrained parameters (Doney *et al.*, 2001). Even the most observable parameters, such as
phytoplankton growth rates, include substantial uncertainty: Laboratory measurements produce considerable scatter around
idealized representations, field observations that loosely constrain model parameters are sparse, and, analytical/instrumental
error exists. The implementation of observed parameters into models can also introduce error; for instance, the extrapolation
of *in situ* bottle samples to a model grid cell having dimensions of kilometres poses a scalability challenge, and often



measurements and models focus on related but different types of information (e.g., measuring pigments but modelling biomass) (Matear and Jones, 2011). In addition, temporal and/or spatial variability in parameter values due to natural variability in phytoplankton species or unrepresented ambient conditions further complicates parameter selection. Identification of the best parameters to use in the model usually requires a lengthy process of fine-tuning often carried out

manually and occasionally in an automated fashion (e.g., Mattern *et al.*, 2016). Both approaches attempt to reproduce available observations of state variables; due to its continuous acquisition and spatial coverage, remotely sensed chlorophyll is the most abundant BGC data set for marine ecosystem model evaluation.

Uncertainty intrinsic to BGC models does not derive from parameter estimation alone but also extends to the choice of equations used to describe the targeted ecosystem (Franks, 2009). These models generally aggregate plankton populations

into broadly defined trophic compartments and track the flow of a chemical element, such as nitrogen or carbon, among these compartments. Variations in this model structure include the use of additional limiting nutrients (such as silicate, phosphate, and iron), the division of the planktonic groups into multiple functional types or size classes, and inclusion of additional state variables such as bacteria or detritus.

Despite their associated uncertainty, BGC models have proven to be exceptional tools. They are currently applied in

research or in support for decision-making. BGC models vary widely in complexity, from simple one-dimensional box models, to global physical-biogeochemical coupled simulations (Doney *et al.*, 2001). Phytoplankton is the first link in the marine food chain and plays an integral role in marine biogeochemical cycling. Therefore, coupled model configurations that attempt to realistically resolve ocean features and their impact on phytoplankton provide invaluable insight for a diverse range of topics including fisheries, water quality and ecosystem health management, carbon sequestration, and climate

change. They provide a mechanistic understanding of the targeted ecosystem and allow to quantify when, where and how changes in phytoplankton distribution and biomass occur.

The East Australian Current (EAC) is the Western Boundary Current of the South Pacific subtropical gyre. It is formed in the South Coral Sea (15 - 24º S - Ridgway and Dunn, 2003) and dominates the large-scale flow along southeastern Australia to the Tasman Sea (24 - 40º S, Fig. 1a). The EAC advects warm oligotrophic waters poleward, displacing cooler

and generally more productive waters, generates mesoscale eddies (Everett *et al.*, 2012), and induces coastal-upwelling (Roughan and Middleton, 2004).

The EAC typically separates from the coast between 30.7°S and 32.4°S (Cetina-Heredia *et al.*, 2014) meandering eastward and leaving a dynamic southward-moving eddy field. In the vicinity and to the north of the separation zone EAC dynamics strongly condition the phytoplankton distribution, while downstream of the separation zone, seasonal effects are

more significant drivers of chlorophyll patterns than upwelling or EAC-derived eddies (Everett *et al.*, 2014). The region's ecosystem is generally nitrogen limited (Hassler *et al.*, 2011) and seasonal increases in nitrate supply during austral winter-spring induce phytoplankton blooms that cover a large area, with particularly large amplitudes on the southern part of the region and around Tasmania (41.6° S, 146.3° E). These blooms are caused by processes that change with latitude, including (a) the contraction of the subtropical gyre over autumn and winter, with the oligotrophic EAC waters replaced by nutrient-



rich subantarctic waters, and (b) nutrient replenishment to the euphotic zone caused by the winter deepening of the mixed layer (Condie and Dunn, 2006).

Outside the annual winter-spring bloom period, phytoplankton variability is linked to mesoscale features (Mongin *et al*., 2011). Pockets of high phytoplankton concentrations south of the EAC separation zone are consistently observed in remote

sensing products and bio-physical models of the region (Baird *et al*., 2006; Macdonald *et al*., 2009). These pockets are associated with persistently elevated concentrations of nitrate in the upper 100 m ($>$ 4 µM, CARS, Ridgway and Dunn, 2003) caused by separation and upwelling events, which have been shown to deliver ten times more nutrients to the shelf than river or sewage discharges (Pritchard *et al*., 2003).

The region's mesoscale eddies are also of special interest. Both cyclonic and anticyclonic eddies are found in large

numbers - with an average of 17 eddies on any given day - and affect surface chlorophyll concentrations (Everett *et al*., 2012). Their contrasting (cyclonic/anti-cyclonic) dynamical regimes create different biogeochemical environments: cyclonic eddies present low sea level anomalies, doming isopycnals and a shoaling nutricline, while anticyclonic eddies are associated with high sea level anomalies, isopycnal depression and a deepening nutricline (McGillicuddy, 1998). Eddies close to the shelf may entrain biomass-rich shelf waters which are then transported offshore (Tranter *et al*., 1986, Everett *et al*., 2015,

Macdonald *et al.* 2016).

While there have been several efforts to model phytoplankton variability and their mechanisms off southeastern Australia, these have been limited to climate change scenarios at the mesoscale (with a spatial resolution of 1/10º - Matear *et al*., 2013) or process studies (e.g., Baird *et al*., 2006a, 2006b, Macdonald *et al*., 2009 and Laiolo *et al*., 2016) that did not attempt to analyse the dynamics at a regional scale. Here we have coupled an $N_2PZD_2$ BGC model (Fennel *et al*., 2006) to a

three-dimensional regional oceanic circulation model for the EAC System (Kerry *et al*., 2016) to investigate the BGC dynamics of the region and, for the first time, create a mechanistic understanding of the system dynamics as a whole. The validation effort, presented here, focuses on objective model performance assessments through comparison of surface chlorophyll model output with an extensive satellite dataset (Copernicus-GlobColour 4km 8-daily product). We also compare the simulated spatial variability in subsurface nutrient concentration (nitrate) to a climatological dataset (CARS -

CSIRO Atlas of Regional Seas climatology).

## 2 Methods

### 2.1 Physical ocean model

We use the Regional Ocean Modeling System (ROMS, version 3.4) to simulate the circulation of the EAC System for the 2002-2011 period. ROMS is a free-surface, hydrostatic, primitive equation ocean model solved on a curvilinear grid with

a terrain-following vertical coordinate system (Shchepetkin and McWilliams, 2005); it has been successfully used in many regional BGC studies (e.g., California Current System - Powell *et al*., 2006, Fiechter *et al*., 2018; North Pacific - Kishi *et al*., 2007; Middle Atlantic Bight - Fennel *et al*., 2006). We use the 10-year free-running ROMS simulation configured for the



EAC region developed by Kerry *et al.* (2016). While the pertinent details are summarised here, the reader is referred to Kerry *et al.* (2016) for a thorough description and validation of the EAC hydrodynamic model.

The model domain (Fig. 1a) extends from 25.25º S to 41.55º S and nearly 1000 km offshore. The northern boundary is chosen at a latitude where the EAC is clearly defined and upstream of the region of elevated eddy variability (Cetina Heredia *et al.* 2014). The model has a 5 km (1/22º) horizontal resolution in the along-shore direction and gradually varies from 2.5 km (1/44º) resolution over the continental shelf and slope to 6 km (1/18º) in the open ocean. The model grid is rotated 20 degrees clockwise, so that it is oriented predominantly along-shore in the *y* dimension and cross-shore in the *x* dimension. We use the vertical stretching scheme of Souza *et al.* (2014) for the vertical distribution of 30 terrain-following sigma layers which have higher resolution in the upper 500 metres to resolve the wind-driven mesoscale circulation and near the bottom for improved resolution of the bottom boundary layer. The bathymetry was obtained from the 50 m multibeam data set for Australia from Geoscience Australia (Whiteway, 2009), and we use the Mellor and Yamada (1982) level-2.5, second-moment turbulence closure scheme (MY2.5) to parameterise vertical turbulent mixing of momentum and tracers.

The model obtains initial conditions and daily boundary forcing from the BlueLink ReANalysis version 3p5 (BRAN3; Oke *et al.*, 2013). BRAN is a multi-year integration of the Ocean Forecasting Australian Model (OFAM) and the Bluelink Ocean Data Assimilation System (BODAS; Oke *et al.*, 2008). At the open lateral boundaries, the Chapman condition (Chapman, 1985) is applied to the free surface and the Flather condition (Flather, 1976) is applied to the barotropic velocity, to ensure that the barotropic energy is effectively transmitted out of the domain. Atmospheric forcing fields from the National Center for Environmental Prediction (NCEP) reanalysis atmospheric model (Kistler *et al.*, 2001) are applied every 6 h, by computing the surface wind stress and surface net heat and freshwater fluxes using the bulk flux parameterisation of Fairall *et al.* (1996). We run the model with realistic forcing from 2002 to the end of 2011 but allow for a year of model spin-up, using the 2003-2011 period (hereafter referred to as the study period) for the analyses.

### 2.2 Biogeochemical model

We use the BGC model of Fennel *et al.* (2006), which was initially developed to assess nitrogen cycling in the Middle Atlantic Bight (Fig. 1b; see also Fig. 1 of Fennel *et al.*, 2006). The model simulates two groups of Nutrients (Nitrate and Ammonium), one group of producers (Phytoplankton), one group of consumers (Zooplankton) and two groups of differently sized Detritus, in what is usually termed an $N_2PZD_2$ model. The model is based on the Fasham *et al.* (1990) parameterisations with the important addition of phytoplankton chlorophyll (mg chlorophyll m$^{-3}$) as an estimate of the chlorophyll stored in phytoplankton, considering the effects of acclimation in the carbon to chlorophyll ratio (Geider *et al.*, 1997).

In the Fennel *et al.* (2006) model, nitrogen available in inorganic nutrients is incorporated into phytoplankton biomass through phytoplankton growth; it is then moved into zooplankton biomass via grazing or into the small and large detrital pools through mortality. Zooplankton mortality also contributes to the detrital pool. Zooplankton losses due to inefficient



grazing and detritus are transferred via a decay rate into the ammonium group, which is transformed into nitrate through nitrification processes. Large and small detritus, as well as phytoplankton, have an associated vertical sinking rate.

We initialize the model with nitrate ($NO_3$) derived from the CSIRO Atlas of Regional Seas climatology (CARS, described in Sect. 2.3.2) and apply daily boundary forcing interpolated from seasonal values of the CARS nitrate climatology. We nudge the model to CARS nitrate over the first 10 grid points (approximately 50 km) from the northern, eastern and southern boundaries. This nudging avoids spurious phytoplankton growth caused by upwelling at the boundaries, identified in previous model runs by a thin lateral zone of high chlorophyll standing stock near the boundaries. The nudging time decreases linearly from 5 to 30 days, from the outermost grid cell of the domain to the interior.

The main parameters used in this configuration are presented in Table 1. The values are all within the common ranges defined in the literature and are comparable to other model configurations in the region (Macdonald, 2013; Matear *et al.*, 2013). Most values are identical to those used by Fennel *et al.* (2006), with the exceptions of the initial slope of the P-I curve and half-saturation constants for the uptake of nitrate and ammonium (bolded values in Table 1). These have been modified by fine-tuning the model to best fit the offshore chlorophyll concentrations available through remote sensing (described in Sect. 2.3.1).

Shelf phytoplankton species and community structure are expected to be different from the phytoplankton community found offshore (Armbrecht *et al.*, 2013). For this study, we ignored these differences and chose to apply an established, relatively simple biogeochemical model with only one phytoplankton functional type. In part, this decision reflected the overall emphasis of the physical model on the EAC and vast offshore region; the model has limited ability to resolve critical physical dynamics on the shelf due to model and forcing resolution and omitted freshwater inflow. Our overall focus is on the larger scale BGC dynamics, their seasonal variability, and smaller-scale impacts of offshore mesoscale processes. Future modelling efforts will address shelf processes and more complex biogeochemical interactions.

## 2.3 Observational datasets

### 2.3.1 Remotely-sensed surface chlorophyll

Remotely-sensed surface chlorophyll estimates were obtained from the Copernicus Marine Environment Monitoring Service (CMEMS) GlobColour product. This product has a 4 km spatial resolution and is generated by fitting a bio-optical model to the merged set of observed normalised water-leaving radiances from the SeaWifs, MERIS, MODIS-A and VIIRS-N sensors (Maritorena and Siegel, 2005). We use 8-daily composite data in order to minimise the data gaps without substantially compromising the time resolution. This data is re-gridded to our model grid through linear interpolation and gap-filled using the DINEOF package (http://modb.oce.ulg.ac.be/mediawiki/index.php/DINEOF). DINEOF gap-fills by iteratively decomposing the data field via Singular Value Decomposition (SVD) until a best solution is found. This solution is achieved by comparison with the subset of reference values (non-gaps) and by progressively including more EOFs in the reconstruction of missing values until the minimisation of error converges (Alvera-Azcarate *et al.*, 2010).





### 2.3.1 Climatological nitrate observations

We used the CSIRO Atlas of Regional Seas climatology (CARS; Ridgway and Dunn, 2003) as our source for three-dimensional nitrate fields for initial conditions, boundary forcing, and to assess the model's ability to reproduce the vertical nitrate distribution in the model domain. CARS is a gridded atlas of mean and seasonal ocean water properties obtained from a quality-controlled archive of all available historical subsurface measurements of ocean properties, that covers the full global ocean on a 0.5º grid. The nitrate fields, created in June 2011, include the World Ocean Circulation Experiment (WOCE) and World Ocean Database 2009 (WOD09) data sets.

### 2.4 Model evaluation metrics

We use six quantitative metrics to assess model skill (adapted from Stow *et al*. 2008):

RMSE - the root-mean-squared error:

$$RMSE = \sqrt{\frac{\sum_{i=1}^{n}(P_i - O_i)^2}{n}}, \tag{1}$$

AE - the average error (bias):

$$AE = \frac{\sum_{i=1}^{n}(P_i - O_i)^2}{n} = \overline{P} - \overline{O}, \tag{2}$$

AAE - the average absolute error:

$$AAE = \frac{\sum_{i=1}^{n}|P_i - O_i|}{n}, \tag{3}$$

BAMEF - the bias-adjusted modeling efficiency:

$$BAMEF = \frac{\left(\sum_{i=1}^{n}(O_i - \overline{O})^2 - \sum_{i=1}^{n}(P_i - \overline{P} - (O_i - \overline{O}))^2\right)}{\sum_{i=1}^{n}(O_i - \overline{O})^2}, \tag{4}$$

RI - the reliability index:

$$RI = \exp\sqrt{\frac{1}{n}\sum_{i=1}^{n}\left(\log\frac{O_i}{P_i}\right)^2}, \tag{5}$$

P$r$ - the Pearson correlation coefficient:

$$Pr = \frac{\sum_{i=1}^{n}(O_i - \overline{O})(P_i - \overline{P})}{\sqrt{\sum_{i=1}^{n}(O_i - \overline{O})^2 \sum_{i=1}^{n}(P_i - \overline{P})^2}}, \tag{6}$$

where n is the number of satellite chlorophyll observations, $O_i$ is the *i*th of n observations, $P_i$ is the *i*th of *n* model chlorophyll predictions, and $\overline{O}$ and $\overline{P}$ are the satellite observation and model prediction averages, respectively. We calculate these for each surface grid cell of the model in order to generate spatial maps of model skill.



The root mean squared error (RMSE), average error (AE), and average absolute error (AAE) measure the size of the discrepancies between model predictions and the observations. So, the closer their value is to zero, the better the match and hence the model accuracy. The AAE is presented to mitigate the fact that values of AE near zero may be created by positive and negative discrepancies cancelling each other. The bias-adjusted modeling efficiency (BAMEF) quantifies how well a model simulates the observation compared to the average of the observations (Nash and Sutcliffe, 1970; Loague and Green, 1991), considering the overall bias for each point. Values less than zero indicate that the observation average would be a better predictor than the model results, zero indicates that the model predicts individual observations no better than the average of the observations, and a value near one indicates a close match between observations and model. The reliability index (RI) measures the average factor by which model predictions differ from observations (Leggett and Williams, 1981). Ideally, the RI should be close to one. An RI of 2.0, for example, indicates that a model predicts the observations within a multiplicative factor of two. Lastly, the Pearson correlation coefficient (P$r$) is a measure of the strength of the linear association between model and observations. It varies between -1 and 1, with negative values indicating that observations and model vary inversely, a value of zero indicates the model and observation variability is not related, and positive values showing that the model varies with the observations (Stow *et al*. 2008).

To assess how well the model is able to represent the main chlorophyll distribution patterns, we determined the dominant orthogonal spatial and temporal signals in the model output and compared them with those of the satellite estimates through an Empirical Orthogonal Function (EOF) analysis (Bjornsson and Venegas, 1997). We used detrended 8-daily data over the full study period and excluded the points between the coast and shelf break (identified as the 200 m isobath). This exclusion was created because a small number of outliers on the narrow shelf were dominating the first EOF modes in the satellite estimates. Our model is not expected to reproduce these shelf estimates (as discussed in Sect. 2.2), some of which could be spurious data (Sect. 2.3.1).

## 3 Model evaluation

Here we characterize the model's ability to reproduce the observed surface chlorophyll concentrations for the study period. We focus on temporal and spatial variability, specifically consecutive seasonal cycles and the typical latitudinal gradient. An objective evaluation of model performance is achieved through comparison with satellite-derived surface chlorophyll data using the statistical metrics described in the previous section, and we identify the main spatial and temporal patterns of chlorophyll variability off-shelf. Finally, the vertical distribution of nitrate is validated against CARS climatological values along a latitudinal transect through the model domain and at three different locations.

### 3.1 Variability of surface chlorophyll concentrations

The time series in Fig. 2a illustrates the domain-averaged surface chlorophyll concentrations of both model (blue) and satellite estimates (green). In addition, we investigate the results for 3 latitudinal zones based on chlorophyll regimes



described by Everett *et al.* (2014) and identified in Fig.1a: Northern Zone (NZ) - upstream of the EAC separation zone, from the northern boundary to 30.5° S; Central Zone (CZ) - EAC separation zone, 30.5° S to 33.5° S; and, Southern Zone (SZ) – downstream of the separation, 33.5° S to the southern boundary. The area-averaged time series for each zone are presented in Fig. 2b, Fig. 2c and Fig. 2d, respectively. All the spatially averaged chlorophyll concentrations reveal a distinct annual cycle

and low interannual variability (Fig. 2). Seasonal variations are largest in SZ, and concentrations are highest in all three zones during spring (spring bloom). The spring bloom signature is emphasized in our area-averaged time series due to the broad spatial coverage of this event.

All three zones have a minimum concentration of 0.1 mg chl m$^{-3}$ occurring around the end of the austral summer. Both maximum and minimum chlorophyll values are within the bounds of observed concentrations in this region (e.g., Everett *et*

*al.*, 2014; Matear *et al.*, 2013). The model is able to competently reproduce the timing of the main chlorophyll fluctuations, as denoted by the high correlation coefficients between simulated and satellite-derived chlorophyll timeseries (*r* of 0.76 for the domain-wide average, 0.73 for NZ and CZ and 0.75 for SZ). However, the model overestimates chlorophyll by about 0.2 mg chl m$^{-3}$ in both NZ and CZ (discussed in Sect. 3.2).

Latitudinal gradients in chlorophyll concentration are evident in the temporal mean (Fig. 3). Of special interest is the

signature created by the nutrient-poorer EAC waters, which extends from the edge of the continental shelf to about 2° of longitude offshore and is characterised by lower-than-average chlorophyll concentrations. Generally larger mean values and variances are found on the shelf in observations and to a lesser extent in the model. Overall, the patterns shown by model and observations are similar, with the model slightly overestimating surface chlorophyll offshore and underestimating concentrations near the coast.

Monthly chlorophyll anomalies from the study period reveal a prevalent seasonal cycle in both observations and the model (Fig. 4). Nutrient-poor EAC water impacts the chlorophyll fields as visible by negative anomaly values along the northern and central portions of the coast, relative to offshore values at those longitudes, especially throughout the months of October and November. Finally, we note the signature of warm-core eddies, smaller negative anomalies downstream of the EAC separation latitude (near 35° S), present in both observed and simulated chlorophyll fields (particularly visible in July

and August).

### 3.2 Skill metrics

To further assess model skill, we solve Eqs. (1-6) at each model surface grid cell. The resulting spatial maps are shown in Figure 5. On average, the model overestimates chlorophyll concentrations offshore, and more so in the north than in the south of the domain (RMSE and AE of ~ 0.2 mg chl m$^{-3}$; Fig 5a and Fig. 5b). The AAE shows fairly constant errors of about

0.25 mg chl m$^{-3}$ (Fig. 5c). The fact that the AE near the south is near zero and the AAE is non-zero implies that the error fluctuates about zero, sometimes positive and sometimes negative. This conclusion is supported by the SZ time-series (Fig. 2d) which reveals that Jan/Feb model overestimation is compensated by short periods of larger amplitude underestimation at times around the spring bloom.



The high AAE values on the shelf indicate that the model represents phytoplankton less well near the coast, generally underestimating chlorophyll concentrations here, except for the shelf region between 29° S to 31° S (Fig. 5b). As discussed in Sect. 2.2, our model calibration focused on simulating the chlorophyll concentrations observed offshore; with only one phytoplankton group represented, the model structure itself limits the ability to simultaneously simulate shelf and offshore

communities. Moreover, it is known that remote sensing products tend to underestimate open-ocean chlorophyll in the study region due to a weak relationship between the large-sized phytoplankton and remote-sensing reflectance (Clementson *et al.*, 1998, Laiolo *et al.*, 2018) and overestimate concentrations near the coast due to the presence of suspended sediments and dissolved organic matter (Moore *et al.*, 2007). Since our model generally underestimates chlorophyll concentrations on the shelf and overestimates them offshore, these remote sensing biases aggravate the model misfit to observations.

The area usually occupied by EAC waters is characterized by low chlorophyll concentrations and low chlorophyll variability. The model tendency to overestimate chlorophyll concentrations offshore and its variability leads to a negative BAMEF in the EAC dominated region (Fig. 5d). Moreover, this is the area where the largest average factor by which model chlorophyll differs from satellite observations is found, with an RI of 2.5 to 3 (Fig 5.e). Model data is well correlated with observations, as demonstrated by the correlation coefficient map (Fig. 5f), with higher correlations in the northern zone (~

0.75) and decreasing towards the south (to ~ 0.6). The overall higher error variability and lower correlations found on the southern zone are likely to stem from the higher eddy activity in the area, in conjunction with generally higher background concentrations (Figs. 2 and 3).

The lower model skill within the EAC revealed by large negative BAMEF and positive RI values results from limited model accuracy capturing high frequency temporal variability in this region. Inconsistencies at high frequencies likely derive

from a combination of physical and biological processes. In a free run such as this, dynamical features like the EAC physical position, mesoscale eddies and small scale fronts, are generally offset in space and/or time from those in nature. In addition, the natural biological community is more complex than the modelled $N_2PZD_2$ system, with only one phytoplankton and one zooplankton compartment. Assessment using these particular metrics reveals the challenge: balancing precision (i.e., how well the model fits each satellite value) with overall accuracy. Even when the main trends and general patterns are well

reproduced, small temporal or spatial lags between events registered by satellite and model can lead to large errors in precision. Matching variability on an 8-day time-scale is a high bar for a non-data assimilative ocean model. Indeed, when calculated using monthly average information, the BAMEF is considerably more positive overall and particularly in the EAC region, and the RI is much smaller over the entire domain; both metrics reveal considerable model skill on monthly average time-scales (Figure 6). Our model tuning and evaluation emphasized the model's ability to generate average spatial and

temporal patterns of chlorophyll, rather than on the absolute values of chlorophyll concentration.

### 3.3 Dominant spatial and temporal patterns

One way to explore the dominant spatial and temporal patterns of phytoplankton variability is through EOF analysis. The spatial EOF fields describe each component in terms of its dominant spatial structures (Fig. 7 - middle and bottom



rows), whereas the principal component's time series give the corresponding temporal weightings for each time step (Fig. 7 - top row). Because this model has been developed to reproduce the general chlorophyll patterns offshore, we exclude shelf (< 200 m isobath) variability. The first four EOF modes explain up to 99.8% and 99.7% of the variance in satellite and model data, respectively.

EOF analysis provides a concise, statistical reduction in the data. As such, there is no guarantee that the statistical modes obtained correspond to dynamical processes or particular features of interest. However, often the modes relate to understandable patterns and forcing time-scales that help identify significant, underlying dynamics or features, and comparisons between observed and modelled modes are often revealing. That is the case for this analysis, with both model and satellite data showing comparable patterns. Mode 1 captures the spring bloom, peaking to its annual maximum around

the beginning of October of each year. Mode 2 represents meridional variations across the domain and includes the effect of the seasonal propagation and recession of EAC nutrient-depleted waters in latitude. Oligotrophic EAC waters replace nutrient-richer Tasman Sea waters expansively during the austral summer and then recede during the fall to winter period.

     Mode 3 captures diverse mesoscale activity, such as increased productivity along the southern edge of the EAC after it separates from the coast – especially noticeable in the satellite-derived dataset, around 35º S. Mode 4 contains most of the

chlorophyll surface response to smaller scale eddy activity. Model and satellite estimates show a similar distribution in both positive and negative signatures, suggesting that the overall geographic effect of productivity enhancement by cyclonic structures and hindrance by anticyclones is well resolved in the model even though the specific time and location of eddies is not expected to be. As previously mentioned, eddies are more prolific in the southern area, hence the higher density of their signatures there. To note an apparent larger/smaller area occupied by the negative/positive signatures, which is on par with

the difference in the average radius of anticyclones (larger) versus cyclones (smaller).

     The principal components of model chlorophyll covary with those of satellite estimates with correlation coefficients of 0.79 for mode 1 and 0.64 for mode 2. As these modes contain around 90% of the variance in both model and satellite estimates offshore, these correlations show that the model reproduces the timing of the dominant fluctuations in chlorophyll concentrations over the domain quite well, in agreement with the high correlation coefficients already found for the area-

averaged chlorophyll concentration timeseries. We note that for reasons described above, the small correlation of principal components for the two mesoscale-dominated modes is not surprising for this non-data assimilative model.

**3.4 Vertical distribution of nitrate**

     As a last step in the model validation effort, we investigate how the model represents the vertical distribution of nutrients by comparing the simulated nitrate with climatological values derived from *in situ* sub-surface data (defined in Sect. 2.3.1).

Specifically, we generate a seasonal transect along an arc at ~156º to 158º E, according to the orientation of the model domain and approximately 200 km from the eastern boundary (Fig. 1a, 8), and vertical point profiles of monthly data at 3 specific locations along the transect (Fig. 1a, 9). We have chosen an across latitude transect because of the observed




latitudinal gradient in chlorophyll concentrations and the transect is fairly offshore, minimising the influence of the EAC (centred at 153º E, west of the transect).

The model represents the distinctive latitudinal gradient in nitrate concentration, showing a poleward shoaling of the nitracline (Fig. 8). In addition, the model reproduces the seasonal cycle well, with the nitracline depth varying ~50 m

upwards/downwards between winter/summer. The larger variability in the simulated nitrate fields reflects processes that are averaged over in the climatology. Therefore, this high-resolution model allows one to examine the impacts of mesoscale phenomena in the regional ecosystem, which would not be possible using climatological fields alone.

The vertical profiles in Fig. 9 represent the monthly-averaged nitrate values (over the full study period for the model, in blue, and from the climatological values of CARS, in red), with the shaded areas representing the first standard deviation.

The general agreement between the model and data at these locations is remarkably good (never exceeding 2 mmol N m$^{-3}$), indicating that ecosystem processes in the model are not causing substantial drift from initial conditions over the 10-year model run. The southernmost profile (Fig. 9a) shows highest concentrations and greatest variability closer to the surface (larger shaded area), resulting from the combined effect of the established latitudinal gradient and the increased mesoscale activity in this area. As both transect and vertical profiles are comparable to the climatological values, this allows us to be

confident in the model's ability to solve nutrient dynamics with depth.

## 4 Conclusions

We have developed a high-resolution (2.5 - 5 km horizontal) $N_2PZD_2$ model of the EAC System and evaluated its performance against satellite-derived estimates of surface chlorophyll and sub-surface nutrient data using several skill metrics. The model is able to reproduce the timing and the spatial structure of the dominant patterns of chlorophyll

variability and the vertical distribution of nitrate. The validation effort is robust and highlights the high skill of this model in reproducing the observed chlorophyll patterns in the EAC System, deeming it suitable for further dynamical studies. We anticipate this coupled configuration will be a much-used framework for exploring how regional oceanic features, and associated biogeochemical dynamics, condition ecosystem response. Such understanding is critical to a diverse range of research areas, spanning from marine ecology to climate change. It is of particular interest in the EAC System due to the

high mesoscale eddy activity observed in the region and because BGC dynamics in Western Boundary Currents remain an understudied topic.

## 5 Code and Data availability

Model initial conditions and boundary forcing come from the Bluelink ReANalysis version 3p5 (BRAN3; Oke *et al.*, 2013) for all the physical variables and from CSIRO Atlas of Regional Seas climatology (CARS2009:

http://www.marine.csiro.au/~dunn/cars2009/) for nitrate. Atmospheric forcing is from the National Center for Environmental





Prediction (NCEP) reanalysis atmospheric model (Kistler *et al*., 2001). Remotely-sensed chlorophyll is the Copernicus-GlobColour 8-daily product generated by ACRI-ST, which can be downloaded from the Copernicus Marine Environment Monitoring Service (CMEMS) catalogue. It is available at http://marine.copernicus.eu/services-portfolio/access-to-products/ with ID: OCEANCOLOUR_GLO_CHL_L4_REP_OBSERVATIONS_009_082. Both physical and BGC model output are

saved as daily snapshots and daily averages of three-dimensional fields of ocean physical and biogeochemical properties (sea-level, temperature, salinity, velocities, nitrates, ammonium, phytoplankton, chlorophyll, zooplankton, large and small detritus), every day over the 10-year simulation period (2002–2011). The data and model code are archived at UNSW Sydney and can be made available for research purposes (contact the corresponding author of this paper).

*Acknowledgements*. This research was supported by an Australian Research Council Linkage Project No. LP150100064, and Discovery Project No. DP140102337 to MR. CR was supported by a UNSW postgraduate TFS award.

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

**Table 1: Main biogeochemical model parameters and their range in the literature (Fennel *et al*., 2006). The values that differ from Fennel *et al*. (2006) are presented in bold and the original value is presented in brackets.**

| Parameter | Range | Value | Unit |
|---|---|---|---|
| phytoplankton growth rate at 0ºC | 0.65[a] - 3.0[b,c] | 0.69 | d[-1] |
| half-saturation concentration for uptake of NO3 | 0.007 - 1.5[d] | **1.0** (0.5) | mmol N m[-3] |
| half-saturation concentration for uptake of NH4 | 0.007 - 1.5[d] | **1.0** (0.5) | mmol N m[-3] |
| initial slope of the P-I curve | 0.007 - 0.13[e] | **0.025** (0.125) | molC gChl[-1] (W m[-2])[-1] d[-1] |
| maximum grazing rate | 0.5[f] - 1.0[g] | 0.6 | (mmol N m[-3])[-1] d[-1] |
| half-saturation concentration of phytoplankton ingestion | 0.56 - 0.2[d,h] | 2.0 | (mmol N m[-3])[2] |
| phytoplankton mortality | 0.05-0.2[i] | 0.15 | d[-1] |
| aggregation parameter | 0.1[d] | 0.005 | (mmol N m[-3])[-1] d[-1] |
| maximum chlorophyll to phytoplankton ratio | 0.005 - 0.072[e] | 0.053 | mgChl mgC[-1] |
| assimilation efficiency | see [j] and [k] | 0.75 | dimensionless |
| excretion rate due to basal metabolism | see [j] | 0.1 | d[-1] |
| maximum rate of assimilation related excretion | see [j] | 0.1 | d[-1] |
| zooplankton mortality | 0.05[l] - 0.25[d] | 0.025 | (mmol N m[-3])[-1] d[-1] |
| remineralization rate of suspended detritus | 0.01 - 0.25[j] | 0.03 | d[-1] |
| remineralization rate of large detritus | 0.01 - 0.25[j] | 0.01 | d[-1] |
| maximum nitrification rate | 0.1[d] | 0.05 | d[-1] |
| light intensity at which the inhibition of nitrification is half-saturated | see [m] and [n] | 0.1 | W m[-2] |
| threshold for light-inhibition of nitrification | see [m] and [n] | 0.0095 | W m[-2] |
| sinking velocity of phytoplankton | 0.009[o] - 25[d] | 0.1 | m d[-1] |
| sinking velocity of suspended detritus | 0.009[o] - 25[d] | 0.1 | m d[-1] |
| sinking velocity of larger particles | 0.009[o] - 25[d] | 1.0 | m d[-1] |



[a] Taylor [1988].
[b] Andersen *et al*. [1987].
[c] Note that owing to the temperature dependence for a temperature range from 0º to 20ºC the maximum growth rate in our model varies
from 0.69 to 2.49 d$^{-1}$.
[d] Lima and Doney [2004].
[e] Geider *et al*. [1997].
[f] Wroblewski [1989].
[g] Fasham [1995].
[h] Note that the values were squared to be consistent with the notation of our model.
[i] Taylor *et al*. [1991].
[j] Leonard *et al*. [1999].
[k] Oschlies and Garcon [1999].
[l] Fennel *et al*. [2001].
[m] Olson [1981].
[n] Note that Olson differentiates between the oxidation of ammonium to nitrite and nitrite to nitrate.
[o] Moskilde [1996].

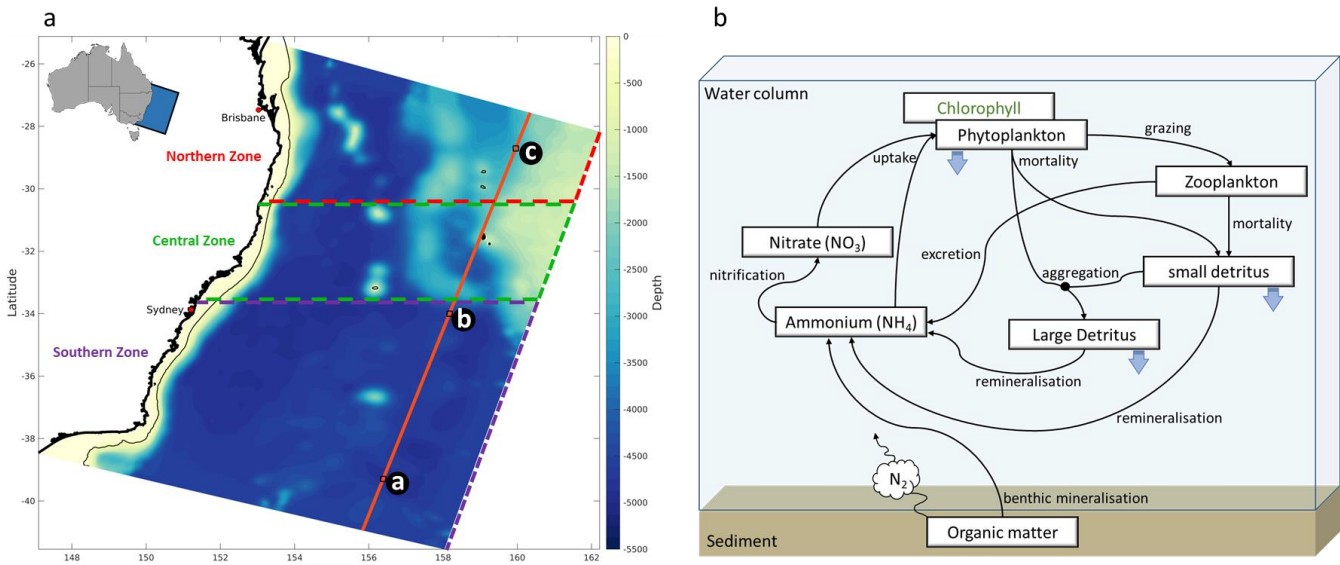

**Figure 1: a) Model domain and bathymetry with the 200 m isobath (thin black line). State capital cities are labelled. Northern Zone extends from northern model boundary to 30.5ºS, Central Zone from 30.5ºS to 33.5ºS and, Southern Zone 33.5ºS to southern boundary. Illustration of the latitudinal nitrates transect (orange line; Fig. 6) and profiles (black squares; letters a, b and c correspond to Figs. 7a, 7b and 7c, respectively). b) Schematic of the biogeochemical model (adapted from Fennel *et al*., 2006).**



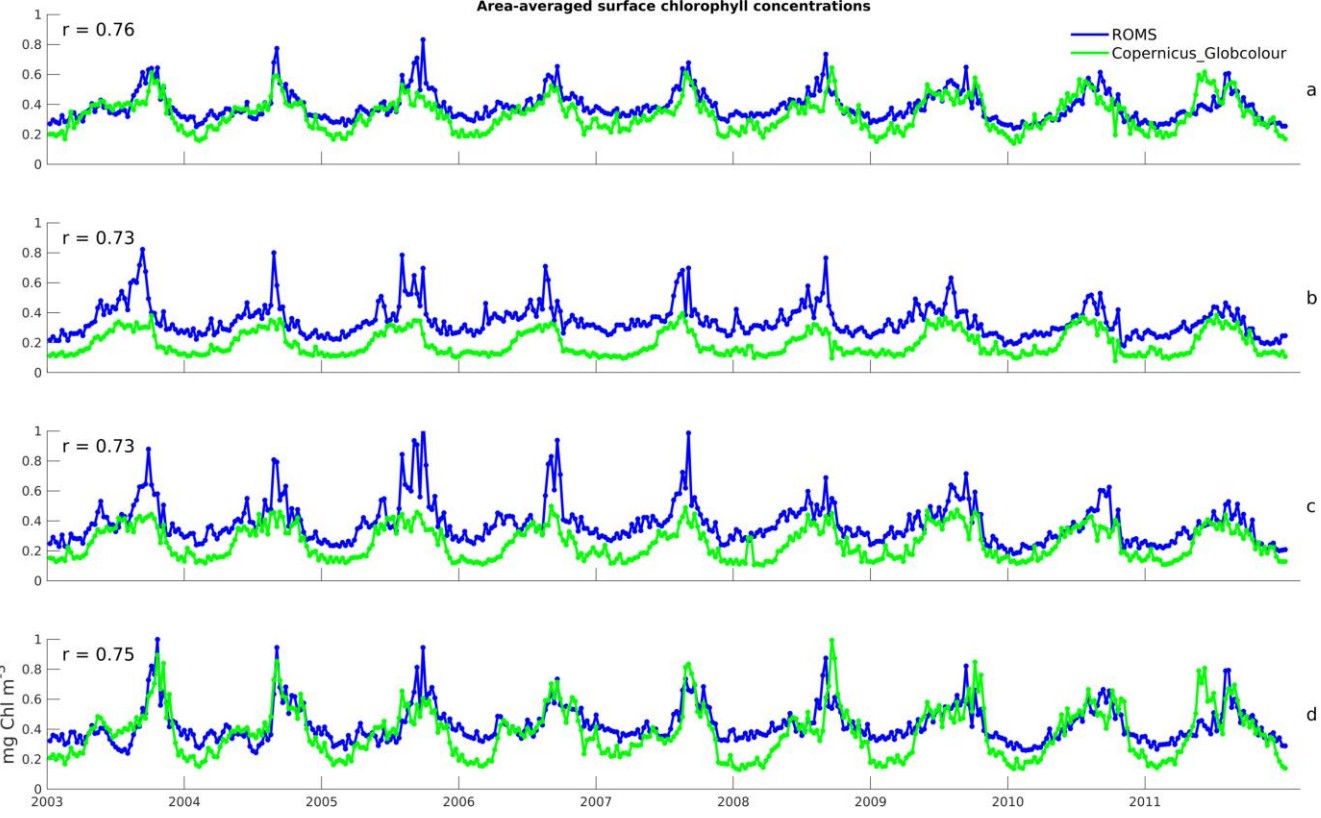

**Figure 2: Time series of area-averaged surface chlorophyll concentrations for 8-daily data from the model (blue) and satellite observations (green): a) Full domain; b) Northern Zone; c) Central Zone; and, d) Southern Zone. The correlation coefficient of the two timeseries, *r*, is shown on the top-left corner of each panel.**





**Figure 3: Surface chlorophyll mean (a,b) and variance (c,d) for the full study period. Satellite observations (left) and model data (right).**



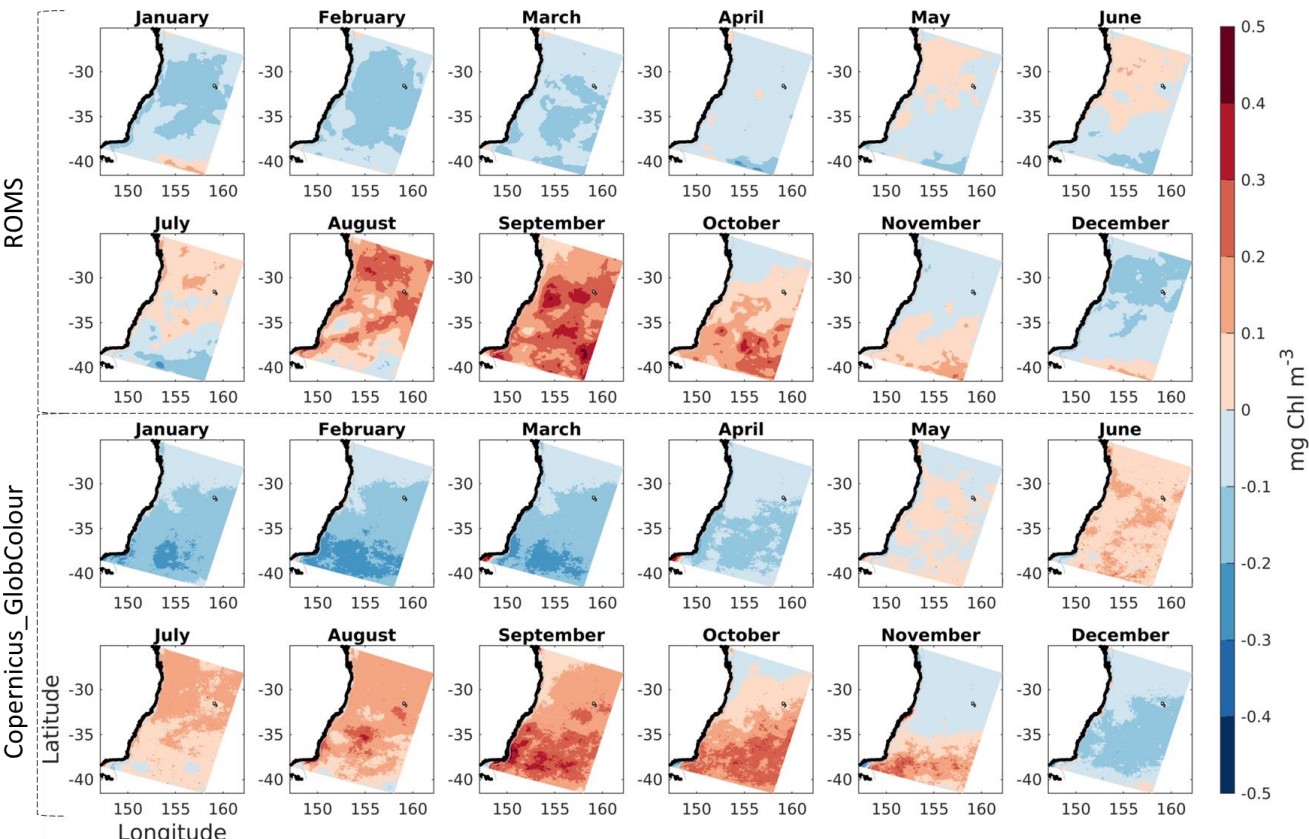

**Figure 4: Monthly anomalies of chlorophyll concentration from model (top two rows) and from satellite observations (bottom two rows).**



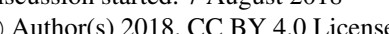



**Figure 5: Maps of a) Root-mean-square error; b) Average error; c) Absolute average error; d) Bias-adjusted modelling efficiency; e) Reliability index; and, f) Pearson correlation coefficient, generated through comparison of simulated 8-daily surface chlorophyll concentrations against 8-daily satellite observations.**

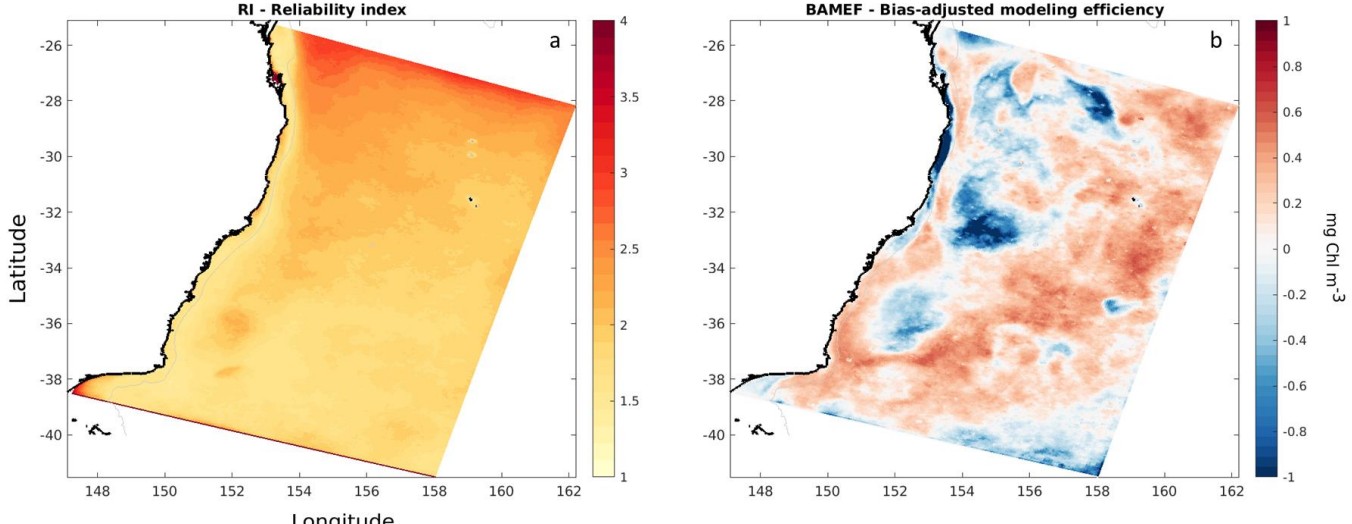

**Figure 6: Maps of a) Reliability index and b) Bias-adjusted modelling efficiency, generated through comparison of simulated monthly surface chlorophyll concentrations against monthly satellite observations.**



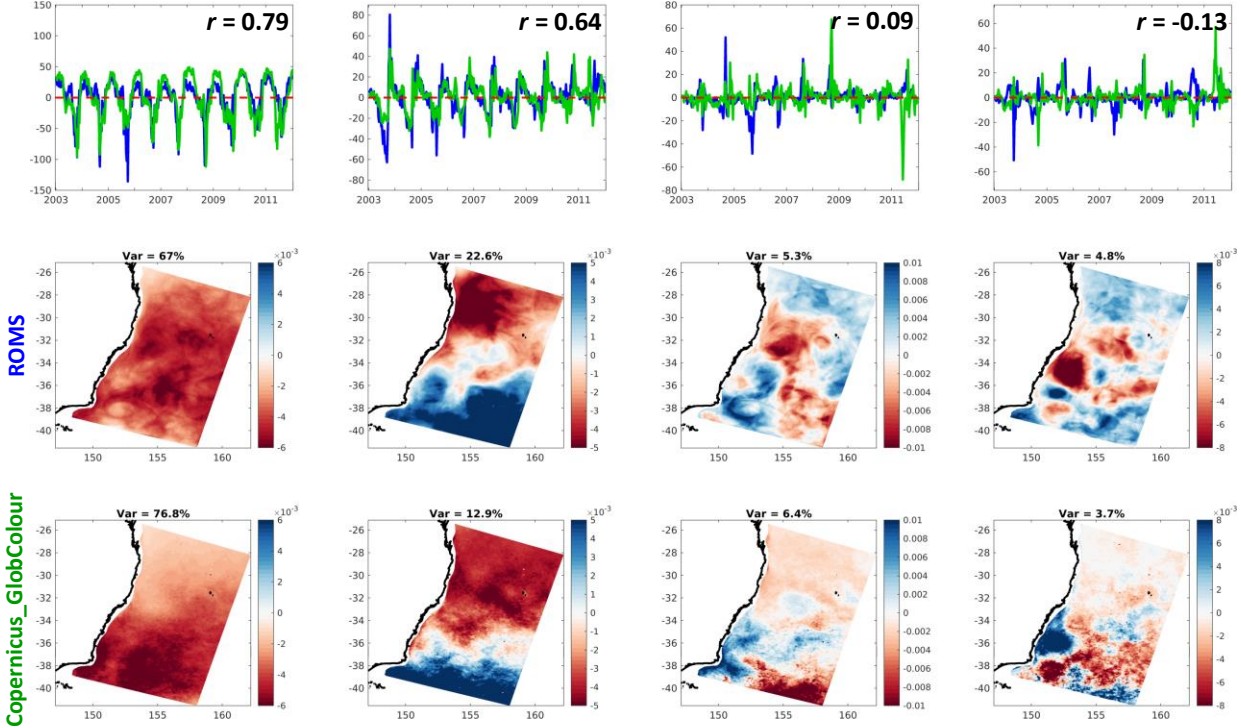

**Figure 7: Principal components (top row: model in blue, satellite observations in green) and spatial structures (middle row: model; bottom row: satellite observations) of the first four EOF modes of surface chlorophyll variance (L to R). The correlation coefficient of the two principal components, *r*, is shown on the top-right corner of each top-row panel.**

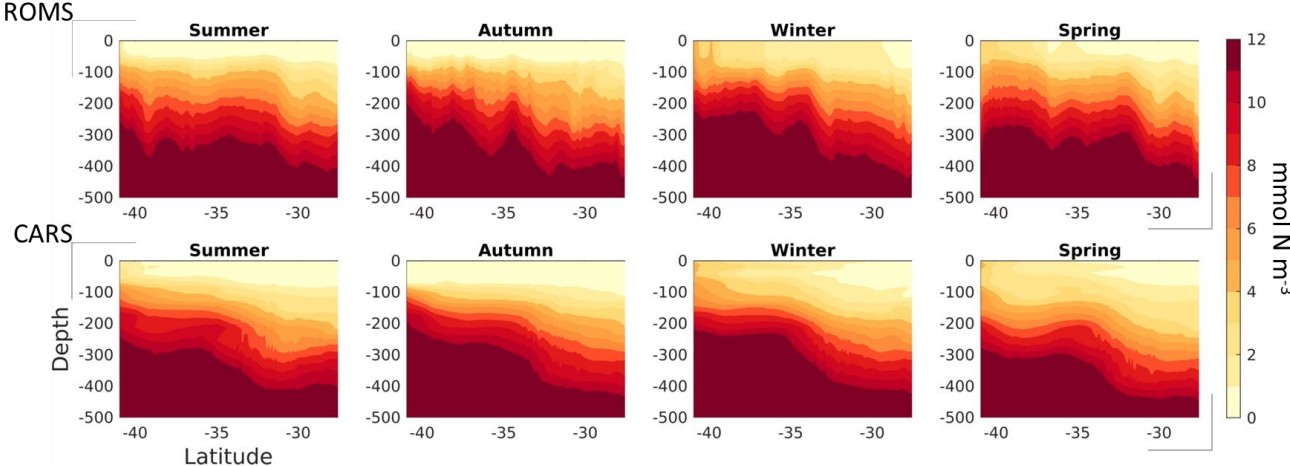

**Figure 8: Seasonal nitrate transects (mmol N m⁻³) along an arc spanning from ~ 156° to 158° E, through the whole domain, South to North (orange line in Fig. 1). Top row: CARS climatological values; Bottom row: model.**





**Figure 9: Monthly nitrate profiles (mmol N m⁻³) at three different locations (a, b and c in Fig. 1). The solid line represents the mean value and the shaded area the first standard deviation. These were calculated over the study period (for model data, in blue) and climatology (for CARS, in red).**

