# Peer review of "A High-resolution Biogeochemical Model (ROMS 3.4 + bio\_Fennel) of the East Australian Current System"

_Geoscientific Model Development, 2018_

## Short Comment (SC1) · 27 Aug 2018

As outlined on https://www.geoscientific-model-development.net/about/manuscript_types.html program code and data need to be made available in an open and persistent way. If this is not possible, e.g. due to copyrights or license issues, reasons need to be stated code availability section. Contacting one of the authors is not seen as persistent form of availability. You may consider to use the DOI service of your university and reference the DOI in your paper.

Lutz Gross GMD Executive Editor

---

## Referee Comment (RC1) · L. Laiolo (Referee) · 10 Sep 2018

General comment

The authors presented a new high-resolution biogeochemical (BGC) model for the East Australian Current (EAC) system. The challenge of this work relies on providing a tool able to:

(1) explore the BGC dynamics in the selected region

(2) understand the EAC system dynamics as a whole

To address these objectives, the authors coupled ROMS and bio_Fennel, obtaining a

model able to explore the complex BGC dynamics of the selected area at a regional and finer scale. The simulated surface chlorophyll-a dynamics were compared with a 10-years dataset of remotely sensed chlorophyll-a product observations (i.e., Copernicus-GlobColour). To assess the model performance several statistical metrics were used. Furthermore, the simulated vertical distribution of the nitrate was assessed against the CARS dataset. The high-resolution model presented here represents a powerful tool to explore the impacts of oceanic features and associated biological responses in the off shore East Australian waters. As stated by the authors in the manuscript text, this would not be possible simply analysing climatological fields by their own. Overall, aims and results of the work are well presented, as well as the different statistical analyses used to assess the simulations. In my opinion, only few sections require clarifications, as detailed below.

Specific comments:

Page 2, line 14: I do not think there is a need to start a new paragraph here.

Page 2, line 27: Same as above, I believe the topic is still the EAC.

Page 3, line 6: Insert the Internet link for CARS

Page 3, line 11- 13: here would be useful to insert the phytoplankton response to these physical factors in both cyclonic and anticyclonic eddies

Page 5, line 25: the link of GlobColour would be helpful for the reader.

Section 2.4: moving the description of the quantitative metrics below every equation can help to better understand the analyses performed and how to read the different panels in Fig. 5 and 6.

Page 8, line 21: EAC nutrient-poor water affect the phytoplankton growth and, as a consequence, the chlorophyll fields.

Page 9, line 8-9: I think the sentence should be reworded. Indeed, rather than 'aggravate the model misfit to observations', the remote sensing biases described earlier can explain the satellite observations vs simulations inconsistencies.

Page 9, line 19-20: I do not understand the meaning of the sentence. The inconsistencies do not derive uniquely from the physic processes, to which corresponds biological responses?

Page 10, line 5-26: the authors here accurately describe the features of figure 7. Therefore, would be helpful to insert mode 1, 2, etc… on the top of Fig. 7, to help the reader to quickly refer to the panels while reading the text.

Figure 1a: It would be useful to shows the EAC and the separation zone and possibly the formation/occurrence of CE and ACE eddies off East Australia. At least a schematic image would be important as I think these information are more relevant for this study rather than the depth.

Figure 8: I imagine the top row is from ROMS and bottom row from CARS. Please double check that, as the figure and caption are not consistent.

———————————————

---

## Referee Comment (RC2) · E. Jones (Referee) · 15 Sep 2018

Summary:

I enjoyed reading this paper and it has the potential of adding to our understanding of BGC dynamics in the EAC separation region. My major criticism at this point relates to the interpretation of the results. I'd like to encourage the authors to consider adding additional interpretation and analysis on a number of fronts that are suggested below. Whilst the comments may appear critical, I think they would strengthen the study.

General Comments:

[Figure]

The initial slope of the P-I curve and half-saturation coefficients have been tuned to recreate the observed Chl-a concentrations. In section 3.1 the model is assessed against observed Chl-a using modelled surface concentrations of Chl-a. The Remotely sensed Chl-a could be considered a some form of "depth weighted" averaged concentration over the optical depth (which can be quite deep in this region). Therefore by taking into account "difference in kind" error between modelled Chl-a and the CMEMS GlobColor Chl-a products, combined with the comparison of a modelled surface Chl-a being compared with a "depth weighted" averaged, there is scope for the "tuning" to be biased. Would it not be better to average the modelled Chl-a over an optical depth?

A majority of the results and discussion focus on Chl-a and Nitrate, yet there are 4 other non-observed state variables that influence the dynamics. What do these distributions look like? Are they sensible? Do they qualitatively behave as one would expect?

There is little discussion of the interaction of physical processes with BGC? For example, the vertical supply of nutrients into the photic zone. What is the typical flushing time of water in the mixed layer? In areas of the domain where the flushing time is short (through horizontal advection), the BGC dynamics will be dominated by the prescribed boundary conditions. Whereas in areas where the flushing time is comparatively long, BGC dynamics will be dominated by internal model processes. Such an analysis would help explain the discrepancies in PCA mode 1 as mentioned below.

The colorbars on many of the figures are such that it is really hard to look quantitatively at the results. It is really difficult to pick discernible differences in color between 0.2 and 0.5 mg Chla m-3. More attention needs to be given to the colormaps used to generate the figures. The addition of a shelf contour to the plots will allow the reader to discriminate the deep ocean, from the shelf and shelf-break.

As it stands the paper is descriptive of observed phenomena, but the power of a model is that it allows you to explore unobservable quantities. There is little if any discussion about the dynamics of the unobserved state variables nor derived quantities like

primary production etc.

If the model is to be used to quantify and interpret the 3D time evolving state of the EAC, then the authors must assess the model in a way that presents evidence to the reader that the model is fit for purpose. Broad statistics are used to show that there is reasonably good correlation between the model and remotely sense observations, but in many cases a detailed interpretation of the results is not presented. Furthermore, the model has only been assessed against remote sensing and in-situ climatology (nitrate), there exists a rich set of BGC observations from gliders and research cruises for the area. I would strongly encourage the authors to undertake an assessment of the BGC model against in-situ data. Why not try a comparison of the model fields against Schaeffer et al., (2016). Whilst it is close to the shelf break, it may assist with providing an additional in-situ dataset for which to assess the model against.

Specific Comments

Page 1, Lines 24-30: There are varying complexities of BGC models ranging from highly parameterised through to extremely complex. The parameter identifiability problem associated with additional complexity is discussed in Friedrichs et al., (2007) with further suggestions on how to adequately represent uncertainty in Parslow et al., (2013). As for using Chl-a as a variable to assess model skill, Baird et al., (2016) show that observed OC3M Chl-a from satellites can at times be very different to simulated Chl-a from a model. This is confirmed in Jones et al., (2016). These "difference in kind errors" are important in the interpretation of the results later in the manuscript.

Page 2, Lines 14 - 21: References needed.

Page 3, Lines 16 - 25: Can you comment as to the suitability of this N2PZD2 model for this particular area? There are other choices available, both more complex and simpler. Is a single P group suitable for this region?

Page 6, Lines 1-8: You mention here that the model is initialised with Nitrate from

CARS. How are the other model variables initialised, especially those that are unobserved?

Page 8, Line 12: What is the likely cause for the bias? It appears that the model is overestimating the Chl-a by a factor of 2 for substantial periods of time. This relates to my question posed above given that you are comparing a surface value with an observed value calculated over an optical depth.

Page 9, Line 3: If you use a 200m depth contour on the plots, it will help denote the region you are discussing.

Page 9, Line 4: Is the model parameterised to simulate large or small phytoplankton?

Figure 7: Suggest adding column titles to denote modes 1-4. Top row - y-axis needs explaining in caption

Page 10, lines 21-26: This section is very light on the analysis and interpretation of the PCA analysis shown in Fig. 7. Whilst the correlation coefficient might be high, the are very obvious differences in the spatial structures of the model and obs. Interpretation is needed to explain these differences beyond just that relating to correlation. e.g. is the model over or under predicting the spring bloom, and in what areas? This may assist in determining why there are discrepancies in the northern section of the domain.

Page 11, lines 1-2: This transect lies so close to the eastern boundary of the model domain that there is a risk that what is being seen in Figure 8 is influenced by the climatological boundary conditions prescribed at the boundaries. What does a transect from the central domain look like?

Figure 9 would benefit from an additional row showing the difference between the model and CARS, such a plot would assist in the interpretation of subtle differences including showing the differences in the supply of nitrate to the surface waters which is important for primary production.

References:

Baird, M.E., Cherukuru, N., Jones, E., Margvelashvili, N., Mongin, M., Oubelkheir, K., Ralph, P.J., Rizwi, F., Robson, B.J., Schroeder, T. and Skerratt, J., 2016. Remote-sensing reflectance and true colour produced by a coupled hydrodynamic, optical, sediment, biogeochemical model of the Great Barrier Reef, Australia: comparison with satellite data. Environmental Modelling & Software, 78, pp.79-96.

Friedrichs, M.A., Dusenberry, J.A., Anderson, L.A., Armstrong, R.A., Chai, F., Christian, J.R., Doney, S.C., Dunne, J., Fujii, M., Hood, R. and McGillicuddy Jr, D.J., 2007. Assessment of skill and portability in regional marine biogeochemical models: Role of multiple planktonic groups. Journal of Geophysical Research: Oceans, 112(C8).

Jones, E.M., Baird, M.E., Mongin, M., Parslow, J., Skerratt, J., Lovell, J., Margvelashvili, N., Matear, R.J., Wild-Allen, K., Robson, B. and Rizwi, F., 2016. Use of remote-sensing reflectance to constrain a data assimilating marine biogeochemical model of the Great Barrier Reef. Biogeosciences, 13(23), pp.6441-6469.

Parslow, J., Cressie, N., Campbell, E.P., Jones, E. and Murray, L., 2013. Bayesian learning and predictability in a stochastic nonlinear dynamical model. Ecological applications, 23(4), pp.679-698.

Schaeffer, A., Roughan, M., Jones, E.M. and White, D., 2016. Physical and biogeochemical spatial scales of variability in the East Australian Current separation from shelf glider measurements. Biogeosciences, 13(6), pp.1967-1975.

---

## Author Comment (AC1) · 15 Nov 2018

We thank the reviewers for their helpful comments and suggestions. We have taken these into careful consideration and changed the manuscript accordingly, which has improved it in content, clarity and presentation. Our responses to the comments are outlined below:

**Short Comment 1 (SC1):**

As outlined on https://www.geoscientific-modeldevelopment.net/about/manuscript\_types.html program code and data need to be made available in an open and persistent way. If this is not possible, e.g. due to copyrights or license issues, reasons need to be stated code availability section. Contacting one of the authors is not seen as persistent form of availability. You may consider to use the DOI service of your university and reference the DOI in your paper.

Thank you for highlighting this. We had forgotten to provide the links to access the model code (for both the physical and biogeochemical components). We have changed the code availability section accordingly. The most current official versions of the code to run the model are made available at: http://www.myroms.org.

**Referee Comment 1 (RC1):**

General comment

The authors presented a new high-resolution biogeochemical (BGC) model for the East Australian Current (EAC) system. The challenge of this work relies on providing a tool able to:

(1) explore the BGC dynamics in the selected region

(2) understand the EAC system dynamics as a whole

To address these objectives, the authors coupled ROMS and bio\_Fennel, obtaining a model able to explore the complex BGC dynamics of the selected area at a regional and finer scale. The simulated surface chlorophyll-a dynamics were compared with a 10-years dataset of remotely sensed chlorophyll-a product observations (i.e., Copernicus-GlobColour). To assess the model performance several statistical metrics were used.

Furthermore, the simulated vertical distribution of the nitrate was assessed against the CARS dataset. The high-resolution model presented here represents a powerful tool to explore the impacts of oceanic features and associated biological responses in the off shore East Australian waters. As stated by the authors in the manuscript text, this would not be possible simply analysing climatological fields by their own. Overall, aims and results of the work are well presented, as well as the different statistical analyses used to assess the simulations. In my opinion, only few sections require clarifications, as detailed below.

**Specific comments:**

Page 2, line 14: I do not think there is a need to start a new paragraph here. Page 2, line 27: Same as above, I believe the topic is still the EAC. We agree, and both have been modified accordingly.

Page 3, line 6: Insert the Internet link for CARS We refer to this in the Code and Data availability section (http://www.marine.csiro.au/~dunn/cars2009/).

**Page 3, line 11- 13: here would be useful to insert the phytoplankton response to these physical factors in both cyclonic and anticyclonic eddies**

Added the information that cyclonic/anticyclonic eddies are usually associated with high/low chlorophyll concentrations (page 3, line 15-20 of the revised version):

"Their contrasting (cyclonic/anticyclonic) dynamical regimes create different biogeochemical environments: cyclonic eddies present low sea level anomalies, doming isopycnals and a shoaling nutricline, while anticyclonic eddies are associated with high sea level anomalies, isopycnal depression and a deepening nutricline (McGillicuddy, 1998). Cyclonic eddies are usually associated with elevated chlorophyll, while anticyclones present chlorophyll suppression (Everett *et al.*, 2012, Gaube *et al.*, 2014). Eddies close to the shelf may entrain biomass-rich shelf waters which are then transported offshore (Tranter et al., 1986, Everett et al., 2015, Macdonald et al. 2016)."

Page 5, line 25: the link of GlobColour would be helpful for the reader.

This is provided in the Code and Data availability section: http://marine.copernicus.eu/services-portfolio/access-to-products/; ID: OCEANCOLOUR\_GLO\_CHL\_L4\_REP\_OBSERVATIONS\_009\_082.

Section 2.4: moving the description of the quantitative metrics below every equation can help to better understand the analyses performed and how to read the different panels in Fig. 5 and 6.

This is a great suggestion for better readability, thank you. The section 2.4 has been modified accordingly.

Page 8, line 21: EAC nutrient-poor water affect the phytoplankton growth and, as a consequence, the chlorophyll fields.

Modified from "Nutrient-poor EAC water impacts the chlorophyll fields as visible by (...)" to "Nutrient-poor EAC water impacts phytoplankton growth, and associated chlorophyll fields, as visible by(...)".

Page 9, line 8-9: I think the sentence should be reworded. Indeed, rather than 'aggravate the model misfit to observations', the remote sensing biases described earlier can explain the satellite observations vs simulations inconsistencies.

Rephrased to "these remote sensing biases can partly explain the inconsistencies between simulated and observed chlorophyll" (page 9, line 25 of revised version).

Page 9, line 19-20: I do not understand the meaning of the sentence. The inconsistencies do not derive uniquely from the physic processes, to which corresponds biological responses?

Both parts (physics and biology) play their role in creating these inconsistencies. Partly because the physical structures are not in the right place at the right time (as it is a free running model), but also because the biological model is an oversimplification of reality, not capturing its complexity, and so it is unable to fully reproduce the observed variability. We have edited the text as follows to clarify this point:

"Inconsistencies at high frequencies likely derive from a combination of physical and biological processes. In a free run such as this, dynamical features like the EAC physical position, mesoscale eddies and small-scale fronts, are generally offset in space and/or time from those in nature. In addition, the modelled  $N_2PZD_2$  system, with only one phytoplankton and one zooplankton compartment, is an oversimplification of reality. By. not capturing the complexity of the natural biological community, it is unable to fully reproduce the observed variability" (page 10, lines 4-9 of revised version).

Page 10, line 5-26: the authors here accurately describe the features of figure 7. Therefore, would be helpful to insert mode 1, 2, etc... on the top of Fig. 7, to help the reader to quickly refer to the panels while reading the text.

Thank you, Fig. 7 has been modified accordingly.

Figure 1a: It would be useful to shows the EAC and the separation zone and possibly the formation/occurrence of CE and ACE eddies off East Australia. At least a schematic image

would be important as I think these information are more relevant for this study rather than the depth.

We have modified Fig.1a to represent the model domain through an 8-day average of the simulated surface chlorophyll (from mid-October 2008), where we highlight the EAC position, one cyclonic eddy and one anticyclonic eddy. We highly appreciate the suggestion as it led to a much more relevant illustration. Thank you.

Figure 8: I imagine the top row is from ROMS and bottom row from CARS. Please double check that, as the figure and caption are not consistent.

Yes, that was a mistake, thank you for detecting it. Corrected.

**Referee Comment 2 (RC2):**

Summary:

I enjoyed reading this paper and it has the potential of adding to our understanding of BGC dynamics in the EAC separation region. My major criticism at this point relates to the interpretation of the results. I'd like to encourage the authors to consider adding additional interpretation and analysis on a number of fronts that are suggested below. Whilst the comments may appear critical, I think they would strengthen the study. General Comments:

The initial slope of the P-I curve and half-saturation coefficients have been tuned to recreate the observed ChI-a concentrations. In section 3.1 the model is assessed against observed ChI-a using modelled surface concentrations of ChI-a. The Remotely sensed ChI-a could be considered a some form of "depth weighted" averaged concentration over the optical depth (which can be quite deep in this region). Therefore by taking into account "difference in kind" error between modelled ChI-a and the CMEMS GlobColor ChI-a products, combined with the comparison of a modelled surface ChI-a being compared with a "depth weighted" averaged, there is scope for the "tuning" to be biased. Would it not be better to average the modelled ChI-a over an optical depth?

This is a great point and one that we had considered. We could have attempted to identify an optical depth based on what the satellite "samples", by using the diffuse attenuation coefficient at 490 nm (1/Kd490), for instance, and then integrating the Chl-a variable over such depth. It is also possible that doing the inverse - converting the 3D Chl-a variable into a 2D field - would be a better approach. The 2D ocean colour product could be determined from the 3D field using (Gordon and Clark, 1980):

$$C_{2D} = \frac{\int_0^z C(z)f(z)dz}{\int_0^z f(z)dz} \qquad \qquad f(z) = exp\left(-\int_0^z 2K_d dz\right)$$

where f(z) is the exponential weighting function that accounts for the arriving irradiance having been attenuated differentially at each depth and returned to the surface by the same factor. This approach has been used to calculate remote-sensing reflectance from depthresolved model inherent optical properties (IOP) fields (Baird et al., 2016) and depthresolved chlorophyll fields (Moline and Prezelin, 2000). However, such approach is also not without limitations - particularly in assuming that the downwelling and upwelling path-lengths are the same.

It is also worth noting that our model covers quite a large area, occupied with different water masses defined by widely contrasting optical properties, which makes the definition of an optical depth far from trivial. Taking all of this into account, we have decided to use the simplest and most common approach (eg. Matear et al., 2013, Cetina-Heredia et al, 2017),

which is to assume that the surface model field is equivalent to the ocean colour product and allows for a direct comparison of the two.

Thus, for these reasons our results remain unchanged.

**A majority of the results and discussion focus on Chl-a and Nitrate, yet there are 4 other non-observed state variables that influence the dynamics. What do these distributions look like? Are they sensible? Do they qualitatively behave as one would expect?**

We have focussed on Chl-a and Nitrate as our interest is in realistically simulating the region's phytoplankton variability. We see the other state variables acting as "closure terms" in helping us reach that goal. However, we qualitatively verified them throughout the model calibration stage. Please refer to the surface means of the remaining state variables, as well and their domain-averaged surface time series, bellow:

Figure S1: Daily surface fields of: a) Zooplankton; b) Ammonium; c) Small detritus, d) Large detritus, averaged over the full study period.